# Heterologous Biosynthesis of Artemisinin in *Chrysanthemum morifolium* Ramat

**Aleksey Firsov [1,\*], Alexander Pushin [1], Svetlana Motyleva [2], Svetlana Pigoleva [1], Lyubov Shaloiko [1], Alexander Vainstein [3] and Sergey Dolgov [1]**

[1]   Branch of the Shemyakin-Ovchinnikov Institute of Bioorganic Chemistry of the Russian Academy of Sciences, Prospekt Nauki, 6, Pushchino, 142290 Moscow Region, Russia; aspushin@gmail.com (A.P.); svoil@rambler.ru (S.P.); shaloiko@yandex.ru (L.S.); dolgov@bibch.ru (S.D.)

[2]   Federal Scientific Selection and Technology Center for Horticulture and Nursery, Zagoryevskaya St. 4, 115598 Moscow, Russia; motyleva_svetlana@mail.ru

[3]   Robert H. Smith Faculty of Agriculture, Food and Environment, The Hebrew University of Jerusalem, POB 12, Rehovot 76100, Israel; alexander.vainstein@mail.huji.ac.il

\*   Correspondence: aleksey_firsov@mail.ru

**Abstract:** Artemisinin-based drugs are the most effective medicine against multidrug-resistant *Plasmodium* spp., the parasite that causes malaria. To this day, wormwood *A. annua* L. is the sole commercial source of artemisinin, where it is produced in minor amounts. The artemisinin yield depends on numerous poorly regulated agricultural factors and the genetic variability of this non-domesticated plant. This has aroused significant interest in the development of heterologous expression platforms for artemisinin production. Previously, we obtained lines of *Chrysanthemum morifolium* Ramat. (*C. morifolium* Ramat.), cvs. White Snowdon and Egyptianka, transformed with artemisinin biosynthesis genes. Here, we report the results of an analysis of artemisinin production in transgenic chrysanthemums. Transcription of heterologous amorpha-4,11-diene monooxygenase and cytochrome P450 reductase genes in transgenic lines was confirmed using high-resolution melting analysis. Artemisinin accumulation was detected using GC-MS in White Snowdon plants, but not in Egyptianka ones, thereby demonstrating the possibility of transplanting active artemisinin biosynthetic pathway into chrysanthemum. Ways of increasing its content in producer plants are discussed.

**Keywords:** plant biotechnology; malaria; heterologous biosynthesis; artemisinin; *Chrysanthemum morifolium*; plant metabolic engineering





## 1. Introduction

Artemisinin, a terpene lactone found in wormwood *Artemisia annua* L. (*A. annua* L.), is currently the most effective drug for treating malaria. The use of drugs combined with artemisinin is recommended by the WHO resolution and is the "gold" standard of treatment worldwide. In addition to malaria, artemisinin and its derivatives are effective against several parasitic helminths, including schistosomes, as well as many human tumor cell lines and some viruses [1]. The primery factor limiting artemisinin use in medicine is its high cost. The content of artemisinin in *A. annua* L. is relatively low, in the range of 0.01–0.8% of dry weight. In addition, the yield of artemisinin varies greatly from year to year, depending on the growth location and on the weather conditions. Improvement of agricultural techniques and breeding of *A. annua* L. has not yet led to a significant improvement in the artemisinin yield. These factors have caused considerable interest in the development of methods for producing artemisinin in heterologous plants, and the artemisinin biosynthesis pathway in *A. annua* L. has been extensively studied to date (Figure S1) [2–4].

Most of the studies on recombinant artemisinin production were carried out using the tobacco species *Nicotiana tabacum* L. and *Nicotiana benthamiana* Domin [5–9]. Only in

the study [10] was *Physcomitrella patens* (Hedw.) Bruch & Schimp. moss used to obtain artemisinin. Although these studies confirmed the possibility of artemisinin production in heterologous plants, its accumulation in the tissues of the host plants was not high. Therefore, the study of the possibility of using plants other than tobacco for recombinant artemisinin production is of considerable interest.

The chrysanthemum is a promising object for research in the field of plant metabolic engineering, including the recombinant artemisinin production. The chrysanthemum belongs to a species rich in terpenoids, sesquiterpenoids and their precursors [11–13]. The high content of various terpenoids is also characteristic for the widespread *C. morifolium* Ramat [14,15]. Chrysanthemum plants stably transformed with genes of the artemisinin biosynthesis pathway—mtADS (amorpha-4,11-diene synthase), CYP71AV1 (amorpha-4,11-diene monooxygenase), CPR (cytochrome P450 reductase), DBR2 (artemisinic aldehyde delta-11(13) reductase) from *A. annua* L. and tHMGR (3-hydroxy-3-methylglutarylcoenzyme A reductase from yeast)—were obtained earlier [16]. The aims of this study were to analyze recombinant artemisinin and explore the features of heterologous CYP71AV1 and CPR genes transcription in transgenic chrysanthemums. Here, for the first time we report on artemisinin detection using GC-MS in chrysanthemums and discuss ways to increase its accumulation.

## 2. Materials and Methods

### 2.1. Plant Material

Chrysanthemums cvs. White Snowdon and Egyptianka were transformed with the p1240 vector as previously described [16]. In the p1240 transformation vector, the genes for the artemisinin pathway, including ADS (amorpha-4,11-diene synthase), CYP71AV1 (amorpha-4,11-diene monooxygenase), CPR (cytochrome P450 reductase) and DBR2 (artemisinic aldehyde $\Delta11(13)$ reductase) from *A. annua* L., were cloned [6] (Figure S2). The ADS gene was cloned with the signal sequence of cytochrome C oxidase subunit 4 (COX4) from *Saccharomyces cerevisiae* Meyen ex E.C. Hansen to define the localization of ADS in mitochondria. The vector also contained a truncated and deregulated tHMGR (3-hydroxy-3-methylglutarylcoenzyme A reductase) from yeast to increase the supply of artemisinin precursor FPP from the mevalonate (MVA) pathway. Chrysanthemums were transformed using the agrobacterium-mediated method. Integration of the target genes was confirmed by PCR (Table S1); only lines transformed with all five target genes were used in the experiments. The transgenic lines were cultivated in a greenhouse until further analysis. Plants were grown at 24 °C during the daytime and 22 °C at night, with a photoperiod of 16/8 h.

### 2.2. GC-MS Analysis

The study used chrysanthemum plants cultivated in a greenhouse for 4 months. Samples were collected from plants in the same growth stage prior to budding. For GC-MS, the leaves were collected, dried in a hot air oven at 56 °C for 18 h and powdered. The samples (100 mg) were extracted overnight in 10 mL n-hexane, filtered, evaporated under $N_2$ gas flow, dissolved in 1.0 mL of methanol, and filtered again. GC-MS analysis was performed as described by [17] with modifications. Artemisinin was analyzed using a JMS-Q1050GC chromatograph (JEOL, Tokyo, Japan) with a DB-5HT column 30 m × 0.25 mm × 0.52 μm (Agilent, Santa Clara, CA, USA). The temperature gradient during the analysis was 40–280 °C; oven temperature progressed from 40 to 130 °C at 1 °C min$^{-1}$, from 130 to 200 °C at 2 °C min$^{-1}$, from 200 to 280 °C at 4 °C min$^{-1}$ and was held at 280 °C for 40 min; the temperature of the injector and interface was 250 °C, and that of the ion source was 200 °C. Gas flow (helium) in the column was equal to 2.0 mL/min, and we used the split-flow injection mode, with the sample injected at a volume of 2 μL; split 1:20; the ionization mode used was electronic impact at 70 eV. BSTFA was used for trimethylsilylation for 30 min and the temperature was equal to 100 °C. Scanning was carried out in the range of 200–900 m/z. Artemisinin was identified by retention parameters

and mass spectra of the NIST-5 library (https://chemdata.nist.gov, accessed on 9 October 2020). The all analyses were performed in triplicate.

### 2.3. RT-PCR and High Resolution Melting (HRM) Analysis

Total RNA was purified from the leaves of transgenic and non-transformed control plants using the Aurum Total RNA Fatty and Fibrous Tissue Kit (Bio-Rad, Hercules, CA, USA). The obtained RNA (3 µg) was used to prepare cDNA using RevertAid Reverse Transcriptase (Thermo Scientific, Waltham, MA, USA) as described in the manufacturer's manual. Amplification of CYP71AV1 and CPR genes transcription products followed by melting were performed on a QuantStudio5 amplifier (Thermo Fisher Scientific, Waltham, MA, USA) in real time mode using a qPCRmix-HS SYBR+LowROX mixture (Evrogen Lab, Moscow, Russia). The primer pairs CYP-For and CYP-Rev (CYP71AV1 gene) and CPR-For and CPR-Rev (CPR gene) were used (Table S1). Melting analysis was performed using QuantStudioTM Design & Analysis Software v1.5.0 (Thermo Fisher Scientific, Waltham, MA, USA). HRM analysis was performed two times, in four replicates in each. The significance of differences in sample Tm was analyzed by a one-way analysis of variance (ANOVA) test (Statistica 6.1 software; StatSoft Inc., Tulsa, OK, USA).

### 3. Results

After agrobacteria-mediated transformation with vector p1240, kanamycin-resistant lines of cvs. White Snowdon and Egyptianka were obtained. All five target genes were detected in only one transgenic line of each variety (Figure S3). These lines, W1 (White Snowdon) and E1 (Egyptianka), were used in further studies. Transgenic chrysanthemum plants did not differ morphologically from their non-transformed counterparts. Expression of the artemisinin pathway genes had no effect on the growth of transgenic plants in the greenhouse and on theirs morphological features.

Artemisinin was detected in the chrysanthemum leaves using GC-MS. The spectrum corresponding to artemisinin was detected in the W1 transgenic line (Figure 1) and corresponded to the peak on the GC chromatogram with RT 21:20 min. In our experiment, the main mass-to-charge ratio for artemisinin was 305 $[M+Na]^+$. At the same time, artemisinin was not detected in the E1 transgenic line. Spectra corresponding to artemisinin were not detected in non-transgenic plants of both studied cultivars.

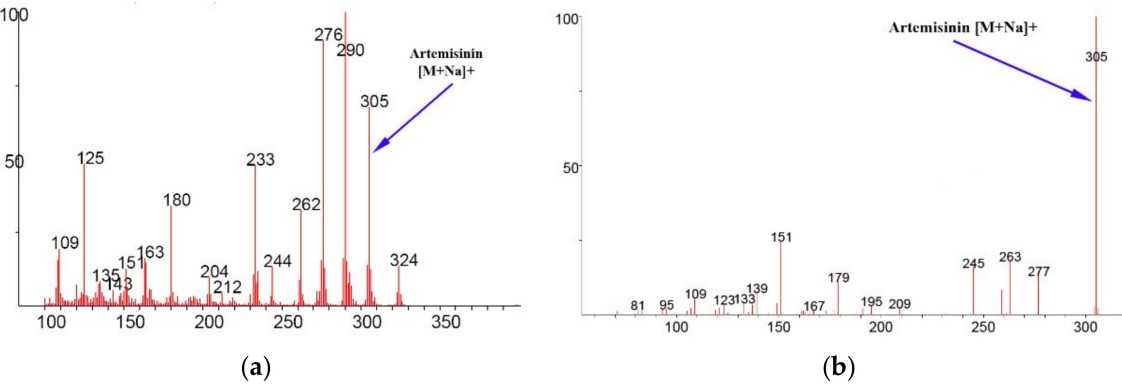

(a)          (b)

**Figure 1.** GC-MS analysis of the samples from transgenic chrysanthemum line W1 (cv. White Snowdon). (**a**) Mass spectrum for peak with RT 21:20 of the sample from transgenic line W1. (**b**) Mass spectrum for the artemisinin standard in the NIST_MSMS library (NIST#: 1166343).

To elucidate the reasons limiting the artemisinin accumulation only to the White Snowdon cultivar, we analyzed the transcription of the artemisinin pathway genes CYP71AV1 and CPR in the leaves of transgenic lines W1 and E1. CYP71AV1, together with its redox partner CPR, converts amorpha-4,11-diene to artemisinic alcohol and further into artemisinic aldehyde (Figure S1). RT-PCR analysis showed amplification of fragments corresponding to genes CYP71AV1 and CPR both in transgenic lines and in control non-

transformed plants (Figure 2). To differentiate the fragments corresponding to the heterologous genes of artemisinin biosynthesis from the chrysanthemum's own sequences, we performed HRM analysis of the RT-PCR products. The analysis results are shown in Figure 2.

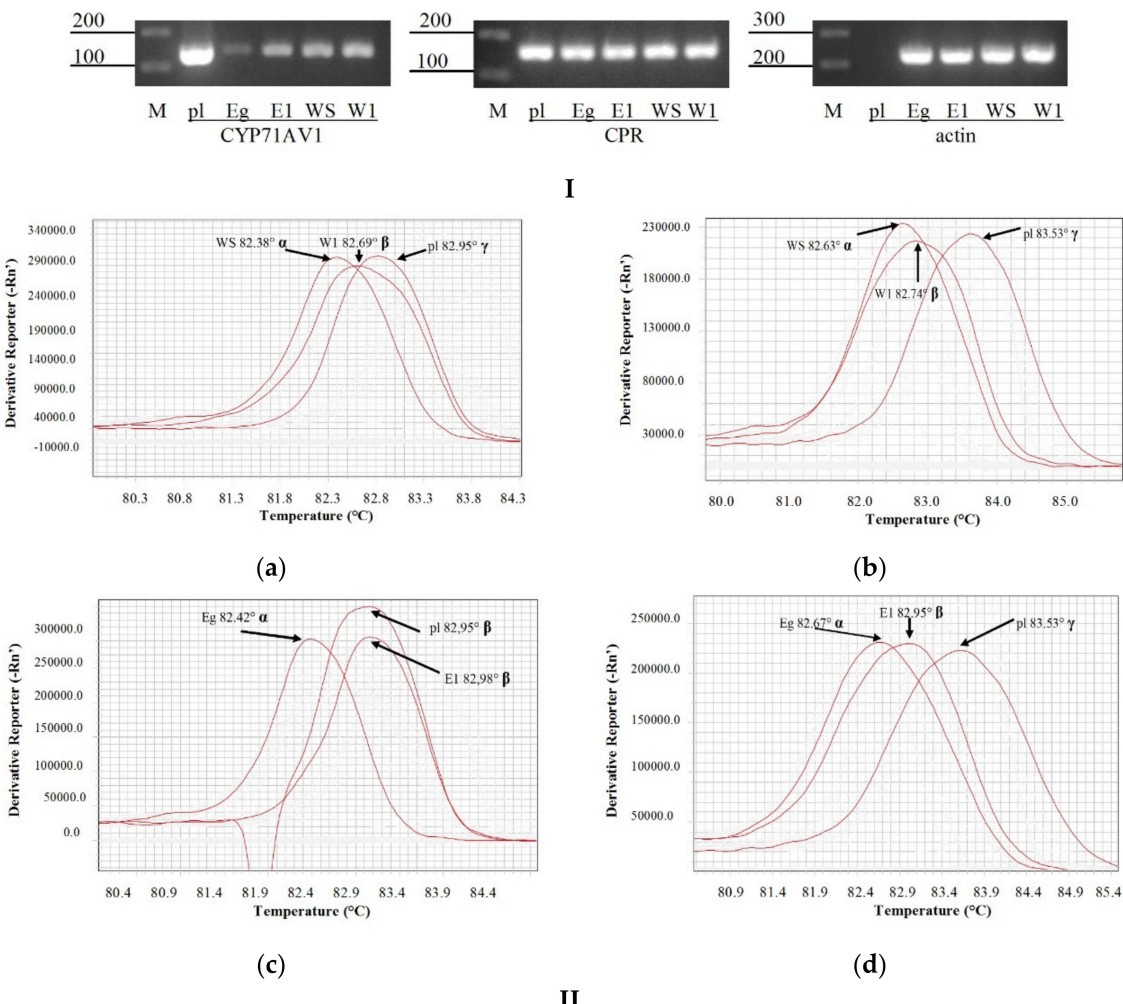

**Figure 2.** RT-PCR and HRM-analysis transgenic and non-transformed control plants. **I**. RT-PCR analysis. pl: DNA of plasmid p1240, positive control; M: molecular size marker, bp. The expected amplified fragments lengths are indicated in Table S1. **II**. HRM-analysis of RT-PCR fragments. (**a**,**b**) cv. White Snowdon; (**c**,**d**) cv. Egyptianka. (**a**,**c**) primers for the CYP71AV1 gene; (**b**,**d**) primers for the CPR gene. WS and Eg: chrysanthemums cvs. White Snowdon and Egyptianka, non-transformed plants, negative control; W1 and E1: transgenic lines cvs. White Snowdon and Egyptianka, respectively. The average Tm of RT-PCR fragments are indicated.

The melting temperature (Tm) of the CYP71AV1 gene fragment from *A. annua* L. was $82.95 \pm 0.12$ °C (plasmid p1240 was used as a template); Tm of the RT-PCR fragment amplified from cDNA of non-transgenic plant cv. White Snowdon was equal to $82.38 \pm 0.01$ °C. At the same time, the Tm of the fragment corresponding to CYP71AV1 amplified from the cDNA of the W1 transgenic line had an intermediate value of $82.69 \pm 0.18$ °C, which indicates the transcription of both the heterologous CYP71AV1 gene and the corresponding sequence from chrysanthemum. In the E1 line, the fragment of the heterologous sequence CYP71AV1 was predominantly transcribed; the Tm of the amplified fragment was $82.98 \pm 0.08$ °C, while the Tm of the corresponding fragment of the non-transgenic plant cv. Egyptianka was $82.42 \pm 0.11$ °C and the Tm of the CYP71AV1 fragment from *A. annua* L. was $82.95 \pm 0.12$ °C.

The Tm of the *A. annua* L. CPR gene fragment in our experiments was equal to 83.53 ± 0.09 °C. The melting temperature of the fragments amplified from cDNA of non-transgenic plants and line W1 (cv. White Snowdon) was 82.63 ± 0.04 and 82.74 ± 0.08 °C, respectively. The differences between these values were significant. The Tm values of the fragments obtained after amplification from cDNA line E1 and non-transgenic plants cv. Egyptianka were 82.95 ± 0.07 and 82.67 ± 0.04 °C, respectively. Thus, HRM analysis confirmed the expression of the heterologous CPR gene in both studied transgenic lines.

## 4. Discussion

In this study, we showed the accumulation of recombinant artemisinin in chrysanthemum line W1, cv. White Snowdon. The presence of artemisinin in the leaf tissues of transgenic plants was confirmed using GC-MS analysis. In another line studied, E1 cv. Egyptianka, recombinant artemisinin was not detected. The both varieties were transformed with the same vector; transcription of artemisinin pathway genes was confirmed in both transgenic lines. We associate the absence of artemisinin in the E1 line with differences in the secondary metabolism processes of the cultivars used in the study. It is known that the biosynthesis of various metabolites and its regulation can differ markedly, not only between chrysanthemum species, but also between varieties within the same species, *C. morifolium* Ramat. [18,19]. This can lead to significant intervarietal differences in the quantity of artemisinin precursors, including a deficiency of its main precursor, farnesyl diphosphate (FPP). Thus, when developing heterologous systems for artemisinin production, it is necessary to take into account the features of the host plant's biochemical background.

In plants, FPP is consumed in numerous biochemical pathways; in particular, it is a precursor of various sesquiterpenes, including being abundantly present in plants' caryophyllene and farnesene. Managing these compounds' biosynthesis can lead to an increase in the availability of FPP for artemisinin synthesis. The FPP content can be increased in various ways, in particular by transferring the heterologous farnesyl diphosphate synthase gene into plants. Current strategies for manipulating IPP supply are summarized in the review [20]. Overall, modifying the metabolic pathways of the heterologous host plants as a way to increase recombinant artemisinin production has not yet been studied in detail.

One of the bottlenecks limiting artemisinin production may be the insufficient expression of artemisinin pathway genes. RT-PCR analysis of non-transgenic chrysanthemums RNA preps showed amplification of the fragments corresponding in size to fragments of wormwood CYP71AV1 and CPR genes. This indirectly indicates the presence in the chrysanthemum genome of sequences similar to the corresponding sequences from *A. annua* L. The presence of such sequences is confirmed by whole-genome sequencing data of the diploid chrysanthemum *Chrysanthemum seticuspe* (Maxim.) Hand.-Mazz. and transcriptome analysis of *C. morifolium* Ramat. [21,22]. HRM analysis of the RT-PCR products confirmed the transcription of the heterologous CYP71AV1 and CPR genes in both studied lines. Based on the Tm curves obtained in the experiments, the fragments corresponding to CYP71AV1 were amplified in the line E1 mainly from the heterologous sequence, and in the line W1 these fragments were from the heterologous and their own sequence equally. At the same time, the fragments corresponding to CPR were amplified mainly from the chrysanthemum's own RNA, especially in the line W1. Cytochrome P450 reductase (CPR) is an electron supply partner of CYP71AV1 and is a core enzyme to synthesize artemisinin [5]. CPR enzyme deficiency may be one of the factors limiting the accumulation of artemisinin in heterologous plants. It should be noted that, unlike other genes of the artemisinin pathway, the role of CPR and its effect on artemisinin accumulation in heterologous plant systems has hardly been studied so far.

## 5. Conclusions

In conclusion, our study clearly demonstrates the feasibility of artemisinin pathway gene expression in nuclear-transformed chrysanthemum plants with consequent accumulation of artemisinin in the plant tissues. Thoughtful selection of chrysanthemum cultivars

with an optimal chemotype as host plants, tuning of competing metabolic branches and the optimization of foreign gene expression cassettes will promote an increase in the accumulation level of recombinant artemisinin, and these are the topics of our future research.

**Supplementary Materials:** The following are available online at https://www.mdpi.com/article/10.3390/separations8060075/s1, Figure S1: Biosynthetic pathway of artemisinin, Figure S2: Schematic depiction of the expression cassette of vector p1240 for artemisinin production in chrysanthemum, Figure S3: PCR analysis of transgenic chrysanthemum lines, Table S1: PCR regimes and nucleotide sequences of used primers.

**Author Contributions:** Conceptualization, A.V., S.D.; in vitro plants cultivation, S.P.; molecular biological, DNA and RNA research, A.F., A.P.; GC-MS analyzes, A.F., L.S., S.M.; project administration, S.D., A.F.; supervision, S.D.; writing—original draft, A.F.; writing—review and editing, A.V. All authors have read and agreed to the published version of the manuscript.

**Funding:** The work was supported by the Russian Science Foundation, Grant No. 19-14-00190 using the unique scientific facility Fitotron reg. #2–2.9.

**Institutional Review Board Statement:** Not applicable.

**Informed Consent Statement:** Not applicable.

**Data Availability Statement:** The data presented in this study are available in this article and its Supplementary Materials.

**Acknowledgments:** We thank A. Blagova for her valuable technical assistance.

**Conflicts of Interest:** The authors declare no conflict of interest.

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
