# Peer review of "Heterologous Biosynthesis of Artemisinin in Chrysanthemum morifolium Ramat"

_separations, doi:10.3390/separations8060075_

Round 1
Reviewer 1 Report
Heterologous biosynthesis of artemisinin in Chrysanthemum morifolium Ramat
Is this article a Brief Report or a research article? In the review report form appears as Brief Report, but in the template o the manuscript as Article. Please decide.
The corresponding author was not mentioned.
Please pay attention to the plant species: first time when they appear should be written with the whole name and between the brackets the short name, for example Chrysanthemum morifolium Ramat (C. morifolium Ramat), and then just the short name can be used. Please put the species name in italic. For example A. annua, but not A. annua. Check this in the whole text.
Line 36 human not “hu-man”
Line 46 species not “spe-cies”
Line 59 reductase not “re-ductase”
Line 61 - accumulation not “ac-cumulation”
Line 62 why is written “2. Materials and Methods” in that place?
Please clearly highlight the aim and originality of the article at the end of the introduction
Figure 1 (especially part B) has low quality. Please provide a better one.
Please describe better the GC-MS method: (1) programed temperature (how it was increasing, hold or how time, etc) (2) was the temperature of injector and ion source negative???; (3) mention the split ratio; describe the derivatization method (procedure, temperature, time, etc)
Regarding the supplementary material, the Arteannuin A and B formulas are not visible (low quality); please provided better quality formulas. The table is not in the journal format. Please see the template.
Reviewer 2 Report
Dear Authors,
Thank you for the nice manuscript. However, your manuscript needs some corrections.
- Please use the italic font for the genus and species name.
- RT 21:20 min or 21.20 min? (Line 73 and 79).
- Please correct the hyphenated words.
- The quality of axis labels dan titles in Figure 2 need to be improved. They are too small and not so clear.
- Unfortunately, there is no information about how much artemisinin from the dried sample was detected. It would be better if the authors could provide this information (if possible).
Best regards,
Reviewer
Reviewer 3 Report
The role of artemisinin in malaria treatment is indisputable. In this purpose artemisinin biosynthesis pathway in A.annua is well studied and various methods for producing arthemisinin are development.
After manuscript reading, one conclusion to Authors it appear. Other concerns were explained in non-published material.
The mechanism of different artemisinin production can be explained of transcriptome level by RNA-Seq or microarray methods. Did you think about such analyzes?
In my opinion, the manuscript is sufficient like a Brief Report after text editing.
Round 2
Reviewer 1 Report
The authors addressed all my comments and suggestions. I suggest that the manuscript will be accepted in the current form.